# Reliability, validity, and correlates of an AI voice emotion recognition app among nurses

Chu-Ying Huang[1], Wen-Pei Chang[2,3]*

1 Taiwan Association of Technological and Economic Cooperation, Taoyuan City, Taiwan, 2 School of Nursing, College of Nursing, Taipei Medical University, Taipei, Taiwan, 3 Department of Nursing, Shuang Ho Hospital, Taipei Medical University, New Taipei City, Taiwan

* 10479@s.tmu.edu.tw

## Abstract

### Background

Digital tools are increasingly widespread in healthcare, particularly in the fields of emotion recognition and mental health assessment.

### Objectives

This study evaluated whether an artificial intelligence (AI) voice emotion recognition (VER) app could identify nurses' emotions and explored its associations with their background and health conditions.

### Methods

The emotions of 349 clinical nurses at a medical center in northern Taiwan were analyzed using an AI VER app and several standardized psychological questionnaires. To control for potential confounding variables, demographic and health-related factors including age, gender, work experience, exercise habits, and history of physical symptoms were collected and statistically adjusted in correlation analyses. Convergent validity was tested with Pearson's correlations, and test-retest reliability was evaluated in 30 nurses using intraclass correlation coefficients (ICCs).

### Results

Significant correlations were observed between app-derived emotions and standard scales (anger: Novaco Anger Inventory-Short Form, $r = .42$; fear: Perceived Stress Scale, $r = .41$; happiness: Oxford Happiness Questionnaire, $r = .45$; and sadness: Beck Depression Inventory-II, $r = .47$; all $p < .001$). Multiple regression identified significant lifestyle and health predictors of emotions: less exercise predicted higher anger ($\beta = .11$, $p = .025$), peptic ulcers predicted greater fear ($\beta = .19$, $p < .001$), daily coffee predicted higher happiness ($\beta = .11$, $p = .041$), and irregular menstrual cycles

**Data availability statement:** All relevant data are within the manuscript and its Supporting Information files.

**Funding:** This research was funded through an industry-academia collaboration agreement between Shuang Ho Hospital, Taipei Medical University, and Bamboo Technology Co., Ltd. in Taiwan, with the project code A-112-006-S. The funders had no role in the study design, data collection and analysis, decision to publish, or preparation of the manuscript. The first author, Chu-Ying Huang, is an employee of Bamboo Technology Co., Ltd. and contributed only to the conceptualization of the study. All other aspects of the research design, data collection and analysis, and manuscript preparation were solely carried out by the corresponding author, Wen-Pei Chang.

**Competing interests:** The authors have declared that no competing interests exist.

predicted lower happiness ($\beta = -.13$, $p = .014$) and greater sadness ($\beta = .30$, $p < .001$). The AI VER app demonstrated good test-retest reliability (ICC = 0.73–0.80).

## Conclusion

Peptic ulcers, irregular menstrual cycles, and lack of exercise were associated with negative emotions such as fear, sadness, and anger. The AI VER app could objectively detect these emotional patterns in nurses, helping to identify emotional fluctuations early and support timely mental healthcare.

## Introduction

Emotions are a crucial factor influencing mental and physical health. For those who work in high-stress environments, such as nurses, emotional management and mental health are even more critical [1]. It has been reported that the heavy workloads and high-stress environments of clinical nurses can exert adverse effects on their emotional state, causing mental health issues such as depression, anxiety, frustration, and hopelessness [2]. Moreover, nurses bear the burden of not only being allocated among different wards and working long shifts but also providing specific patient care based on the conditions of each patient and adapting to varied nursing work in different departments [3].

Technological advances have expanded the use of digital tools in healthcare, particularly for emotion recognition and mental health assessment [4]. Voice emotion recognition (VER) automatically identifies emotional states through speech signal analysis [5]. With rapid progress in speech technology, VER has been applied to complementary medicine and psychological counseling, such as screening for depression, emotional and cognitive disorders, and Alzheimer's disease [6–8]. These applications enhance mental health awareness by enabling timely understanding of emotional states [9]. Designing an accurate and effective VER model remains a key focus of AI-based emotion detection.

Nurses' emotional states may be influenced by factors such as age, gender, and marital status [10]. However, studies report a steady rise in nurses' work-related stress [11,12]. Religious beliefs have also been identified as a protective factor that can help regulate negative emotions in high-stress environments [13,14]. Work stress may lead to physical symptoms like migraines, peptic ulcers, and irregular menstrual cycles [15–17], contributing to fatigue, reduced job satisfaction, and higher turnover, which can impact care quality [18]. Therefore, rapid and accurate emotional assessment may help detect early imbalances and support timely intervention.

AI-based emotion recognition apps can automatically detect users' emotional states, making the evaluation of their reliability and validity essential [19]. However, research on the accuracy of voice emotion recognition (VER) devices remains limited. This study aimed to examine the reliability and validity of an AI VER app among clinical nurses and to explore how their personal backgrounds and physical symptoms were associated with the app's emotion recognition results.

## Materials and methods

### Research design

Clinical nurses aged 20–45 were recruited from a medical center in northern Taiwan between 01/12/2023 and 30/06/2024. Those who were pregnant or taking hormonal drugs were excluded. All voice recordings and questionnaires were collected by a single research nurse to ensure data consistency. AI VER testing was conducted after each nurse's work shift (day, evening, or night) and consisted of two parts: emotion recognition and a questionnaire survey. Voice data were recorded individually in a quiet meeting room using the AI VER app on a smartphone. Each nurse spoke naturally for at least 60 seconds about their clinical experiences or emotions from the previous day, and the AI analyzed four emotions, anger, fear, happiness, and sadness, based on vocal features such as intonation and rhythm. To assess test-retest reliability, 30 of the 349 nurses were randomly selected to repeat the AI VER test three weeks later. The required sample size was estimated using G*Power 3.1 ($\alpha = 0.05$, power = 0.8, effect size = 0.3), with 340 participants accounting for a 15% potential attrition rate.

### Ethical considerations

The VER model used in this study was developed and funded by Bamboo Technology Co., Ltd. All data collection, statistical analyses, and result interpretations were conducted independently by our academic research team at Taipei Medical University to ensure objectivity and prevent developer bias. The company was not involved in data analysis, research procedures, or study conclusions.

This study was approved by the Joint Institutional Review Board of Taipei Medical University (N202309075). Written informed consent was obtained from all participants after explaining the study purpose and procedures. Participation was voluntary, and nurses could withdraw at any time without affecting their rights. All data were de-identified and used solely for academic research.

To protect voice data and participant privacy, recordings were stored on encrypted Microsoft Azure servers accessible only to authorized research team members. During processing, feature extraction retained only acoustic parameters relevant to analysis; original recordings were anonymized, and any identifiable information was removed. Data were labeled with random codes, retained per IRB regulations, and securely destroyed after the approved period. Participants could delete their voice data from the platform at any time.

### Research instruments

The data collection tools used in this study included the basic data and physical symptoms or diseases of the nurses, the Positive and Negative Affect Schedule (PANAS), the Oxford Happiness Questionnaire (OHQ), the Perceived Stress Scale (PSS), the Beck Depression Inventory-II (BDI-II), the Novaco Anger Inventory-Short Form (NAI-25), and our AI VER app. The details are as follows:

(1) **Basic data of nurses:** These included basic attributes such as gender, age, educational background, marital status, religion, years of work experience, department of employment, current shift type, and living habits (including whether they smoked, consumed alcohol or coffee, or exercised).

(2) **Physical symptoms or diseases:** These included physical symptoms such as migraines, peptic ulcers, and irregular menstrual cycles. Migraines were assessed using the International Headache Society's criteria, based on headache frequency, duration, and severity over the past 3–6 months [20]. Peptic ulcer history was determined by self-reported diagnoses or past gastrointestinal bleeding. To assess menstrual cycle regularity, we developed a seven-item questionnaire covering cycle length, duration, flow, and abnormalities [21]. Questions included changes in cycle length or flow, periods longer than 35 or shorter than 24 days, and symptoms such as extreme pain, abnormal bleeding, or

missed periods in the past six months. Any reported abnormality was considered menstrual irregularity. Five experts (gynecologists and nursing professors) evaluated each item for appropriateness and relevance on a 4-point scale. The overall content validity index (CVI) was 94%.

(3) **Positive and Negative Affect Schedule (PANAS):** According to Watson et al. [22], positive affect is linked to social engagement and satisfaction, while negative affect is associated with stress, illness, and poor coping. Positive traits include enthusiasm and alertness, whereas negative traits include anger and fear. This study used the 20-item PANAS to assess nurses' emotional states. The scale contains 10 positive and 10 negative items, rated on a five-point Likert scale (1 = almost never, 5 = always). Higher scores reflect stronger emotional responses. Cronbach's α for positive and negative subscales was 0.86 and 0.87, with test-retest reliability of 0.79 and 0.81, respectively.

(4) **Oxford Happiness Questionnaire (OHQ):** The Oxford Happiness Inventory (OHI), developed by Argyle and Hills, originally measured seven dimensions of happiness but tended to produce inflated scores due to its positive skew [23]. To address this, the Oxford Happiness Questionnaire (OHQ) was later developed. The OHQ, also containing 29 items, was used in this study to assess nurses' perceived happiness. Items were rated on a six-point Likert scale (1 = strongly disagree to 6 = strongly agree), with higher scores indicating greater happiness. The Cronbach's α of internal consistency was 0.84 [24].

(5) **Perceived Stress Scale (PSS):** Developed by Cohen et al., the PSS was used to measure the amount of stress that the clinical nurses perceived and to gauge whether they felt that they lacked control or were unable to cope with things in life. The PSS contains 14 items measured on a five-point Likert scale ranging from never (0 points) to always (4 points), with a higher score indicating a higher level of stress. This scale has good reliability and validity; Cronbach's α of inter-rater consistency was 0.85 [25].

(6) **Beck Depression Inventory-II (BDI-II):** This was co-designed by Beck et al. in 1961 to assess the degree of depression in subjects between the ages of 13 and 80. It has since been frequently used to evaluate symptoms of depression [26]. In 1996, Beck et al. developed the second version, which contains 21 question items, each with four options arranged based on severity and scored from 0 to 3 points. The total score ranges from 0 to 63 points, with 14–19 points indicating mild depression, 20–28 points indicating moderate depression, and 29 points or higher indicating severe depression. For reliability, the test-retest reliability at one week was 0.93, and Cronbach's α of internal consistency was 0.91 [27].

(7) **Novaco Anger Inventory-Short Form (NAI-25):** This was modified from the original NAI and includes 25 items of the original 90 items to more readily assess the degree of anger felt by subjects [28]. Devilly later verified the reliability and validity of the NAI-25, demonstrating that it can effectively detect anger emotions [29]. It had a Cronbach's α of 0.96; the mean inter-item correlation was 0.49; the item-total correlation ranged from 0.50 to 0.77, and the split-half reliability was 0.93.

(8) **AI VER app:** To objectively identify emotional states from voice features, this study adopted an artificial intelligence-based voice emotion recognition (AI VER) system developed for Mandarin speakers in Taiwan. The app combines voice recognition technology, emotion analysis algorithms, and an interactive interface to analyze speech and display emotion feedback. The AI VER app integrates voice recognition, emotional analysis, and a user-friendly interface to detect and visualize users' emotional states. For this study, the Here Hear VER model, developed by Bamboo Technology Co., Ltd. in Taiwan, was adopted (Fig 1). The model analyzes vocal parameters such as intonation, pitch, rhythm, and energy to classify speech segments into four basic emotions: anger, fear, happiness, and sadness. The overall development process of the Here Hear VER model involved three stages: model architecture and training, dataset construction and preprocessing, and model validation and performance testing. Each stage is described below in detail to illustrate how the AI system was developed, optimized, and evaluated for emotion recognition accuracy.

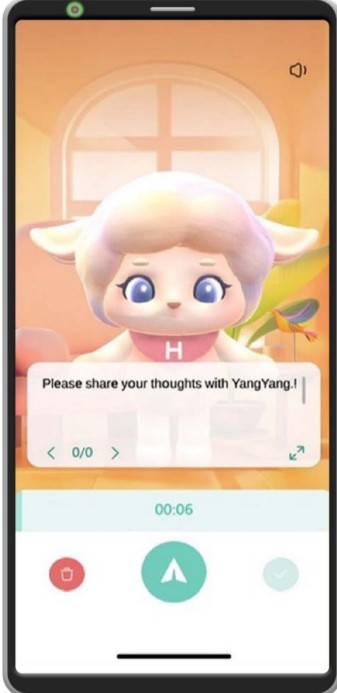 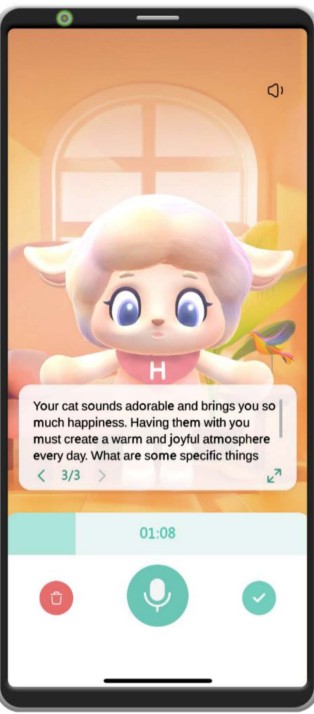 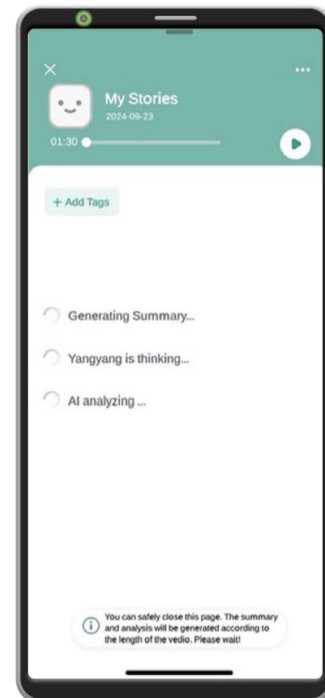 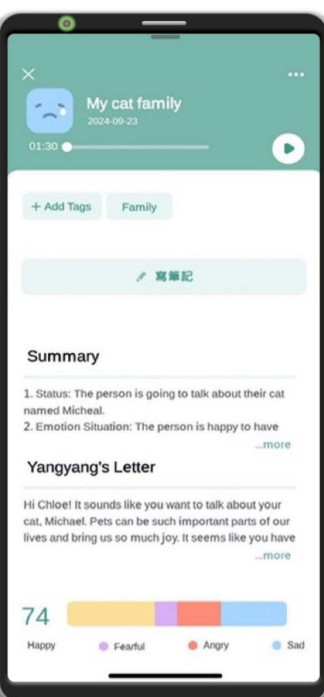

**Fig 1. Example of using the AI VER app, translated from traditional Chinese (the language used in this study) to English.** From left to right: the app beginning interactive recording of the voice of the nurse, the app responding to the nurse's interactive content, the wait for the personalized analysis report to generate, and the emotion analysis report provided to the nurse. (Image provided by Bamboo Technology Co., Ltd., published under a CC BY license, with permission from Bamboo Technology Co., Ltd., original copyright 2025).

**Model architecture and training:** The *Here Hear* model is based on the wav2vec2-Large pre-trained audio representation framework, enabling deep learning of temporal and spectral patterns in speech. The architecture consists of three modules: audio feature extraction, feature selection, and emotion classification. The final classifier is a decision tree model, chosen for its interpretability and practical usability in healthcare contexts. The model extracts emotional features using acoustic characteristics such as Mel Frequency Cepstral Coefficients, pitch, and energy to capture emotional tones in voice [30,31].

**Dataset and preprocessing:** The model was initially trained using the Ryerson Audio-Visual Database of Emotional Speech and Song (RAVDESS), containing 7,356 labeled recordings from 24 professional actors aged 21–40 years. To enhance cultural relevance and performance for Mandarin Chinese speakers, the app was further trained with natural Mandarin voice data collected from users in Taiwan [32]. In total, 13,075 recordings (28,814 minutes) were obtained from 6,457 participants. All recordings underwent preprocessing steps including noise reduction, segmentation, and normalization to ensure consistency. Each voice sample was labeled by three nationally certified counseling psychologists across eight emotion categories (worried, happy, sad, angry, surprised, fearful, calm, and neutral). Multi-label classification was permitted to capture emotional complexity. To ensure consistency, a triangulation method was used, and only data in which at least two psychologists agreed were included. The validated dataset was divided into training, validation, and test sets (8:1:1), achieving an optimized accuracy of 79.8%.

**Model validation and performance:** The Mandarin model demonstrated stable learning progression, with accuracy increasing from 48.9% after the first epoch to 85.9% after the fifteenth epoch. During independent testing, the final model achieved approximately 70% accuracy, indicating moderate classification performance and acceptable consistency.

Emotion proportions were normalized such that A (anger) + B (fear) + C (happiness) + D (sadness) = 1, and the percentage of each emotion was calculated using the following equations [33]:

$$\text{Proportion of anger} = (A/(A + B + C + D)) \times 100\%$$

$$\text{Proportion of fear} = (B/(A + B + C + D)) \times 100\%$$

$$\text{Proportion of happiness} = (C/(A + B + C + D)) \times 100\%$$

$$\text{Proportion of sadness} = (D/(A + B + C + D)) \times 100\%$$

Overall, the integration of deep learning-based feature extraction and decision tree classification enabled interpretable and contextually relevant emotion detection, providing a foundation for reliable emotion assessment in clinical research.

## Statistical analyses

Data were analyzed using the Chinese edition of SPSS 25.0. Nurses' demographic and health-related variables (e.g., age, gender, work experience, department, shift type, migraines, peptic ulcers, menstrual cycle) were checked for completeness and outliers. Descriptive statistics were summarized as frequencies and percentages. Group differences in app-derived emotion scores (anger, fear, happiness, sadness) were examined using independent t-tests or one-way ANOVA, with nonparametric alternatives (Mann-Whitney U or Kruskal-Wallis tests) applied when normality assumptions were violated. Analyses related to menstrual cycles were restricted to female nurses.

Convergent validity was assessed using Pearson's correlation between app-derived emotion scores and established emotional measures, including PANAS, OHQ, PSS, BDI, and NAI. Following Her and Wong [34], correlations below.40 were considered weak,.40–.69 moderate, and ≥.70 strong. To further identify predictors of emotion recognition outcomes and control for potential confounders, multiple linear regression analyses were performed with the four emotion scores as dependent variables and significant background or psychological factors from univariate tests as independent variables.

To evaluate test–retest reliability, 30 nurses completed the AI VER app and related scales twice, three weeks apart. Consistency was examined using Pearson's correlation and intraclass correlation coefficients (ICC). Based on Koo and Li [35], ICC values of 0.50–0.74 indicate moderate, 0.75–0.89 good, and ≥ 0.90 excellent reliability.

## Results

### Background characteristics of the nurses

Data were collected from 349 nurses, most of whom were female (85.4%) and aged 25–29 years (43.6%). The majority had a university degree or higher (77.4%), were single (89.4%), and reported no religious affiliation (59.9%). Most had ≥ 5 years of work experience (40.7%), worked in general wards (63.6%), and were on the day shift (58.7%). Regarding lifestyle, most nurses did not smoke (97.1%) or drink alcohol (65.9%); 38.4% consumed coffee occasionally, and 57.0% did not exercise regularly (Table 1).

### Correlation between background variables and emotion recognition results in nurses

As shown in Table 1, happiness showed the highest mean intensity ($M = 0.44$), while anger, fear, and sadness were lower. Among background variables, only exercise habits were significantly related to anger ($t = 3.14$, $p = .002$); nurses who seldom exercised exhibited more anger. Coffee-drinking habits were associated with happiness ($F = 3.20$, $p = .042$), though no post-hoc difference was found. Other demographic and lifestyle variables showed no significant effects.

**Table 1. Correlation between personal background variables and emotion recognition results of app (*N* = 349).**

| Personal basic attributes | Total (%) | Anger | | Fear | | Happiness | | Sadness | |
|---|---|---|---|---|---|---|---|---|---|
| | | *M ± SD* | *p* | *M ± SD* | *p* | *M ± SD* | *p* | *M ± SD* | *p* |
| | | 0.22 ± 0.20 | | 0.17 ± 0.15 | | 0.44 ± 0.24 | | 0.17 ± 0.18 | |
| Gender | | | .585 | | .488 | | .202 | | .680 |
| Male | 51 (14.6) | 0.21 ± 0.18 | | 0.15 ± 0.15 | | 0.48 ± 0.26 | | 0.16 ± 0.16 | |
| Female | 298 (85.4) | 0.22 ± 0.20 | | 0.17 ± 0.15 | | 0.43 ± 0.24 | | 0.18 ± 0.18 | |
| Age | | | .327 | | .974 | | .384 | | .794 |
| 20–24 years old | 115 (33.0) | 0.24 ± 0.19 | | 0.17 ± 0.15 | | 0.42 ± 0.23 | | 0.18 ± 0.18 | |
| 25–29 years old | 152 (43.6) | 0.20 ± 0.20 | | 0.17 ± 0.16 | | 0.46 ± 0.25 | | 0.17 ± 0.18 | |
| ≥ 30 years old | 82 (23.4) | 0.23 ± 0.21 | | 0.17 ± 0.13 | | 0.43 ± 0.24 | | 0.18 ± 0.19 | |
| Educational background | | | .847 | | .055 | | .745 | | .473 |
| Junior college | 79 (22.6) | 0.22 ± 0.20 | | 0.14 ± 0.13 | | 0.44 ± 0.25 | | 0.19 ± 0.19 | |
| University or higher | 270 (77.4) | 0.22 ± 0.20 | | 0.18 ± 0.16 | | 0.43 ± 0.24 | | 0.17 ± 0.18 | |
| Marital status | | | .841 | | .701 | | .528 | | .837 |
| Married | 37 (10.6) | 0.21 ± 0.24 | | 0.16 ± 0.17 | | 0.46 ± 0.27 | | 0.17 ± 0.19 | |
| Single | 312 (89.4) | 0.22 ± 0.20 | | 0.17 ± 0.15 | | 0.43 ± 0.24 | | 0.17 ± 0.18 | |
| Religion | | | .967 | | .119 | | .161 | | .690 |
| Not religious | 209 (59.9) | 0.22 ± 0.20 | | 0.18 ± 0.16 | | 0.42 ± 0.24 | | 0.18 ± 0.19 | |
| Religious | 140 (40.1) | 0.22 ± 0.21 | | 0.15 ± 0.13 | | 0.46 ± 0.24 | | 0.17 ± 0.17 | |
| Years of work experience | | | .838 | | .226 | | .469 | | .098 |
| Less than 1 year | 39 (11.2) | 0.22 ± 0.17 | | 0.19 ± 0.18 | | 0.39 ± 0.22 | | 0.21 ± 0.18 | |
| 1–3 years | 103 (29.5) | 0.23 ± 0.19 | | 0.18 ± 0.14 | | 0.45 ± 0.23 | | 0.14 ± 0.13 | |
| 3–5 years | 65 (18.6) | 0.20 ± 0.17 | | 0.14 ± 0.09 | | 0.46 ± 0.23 | | 0.20 ± 0.21 | |
| 5 years or more | 142 (40.7) | 0.22 ± 0.22 | | 0.17 ± 0.17 | | 0.43 ± 0.26 | | 0.17 ± 0.19 | |
| Department of employment | | | .646 | | .662 | | .471 | | .070 |
| General ward | 222 (63.6) | 0.22 ± 0.20 | | 0.17 ± 0.14 | | 0.43 ± 0.24 | | 0.19 ± 0.20 | |
| ICU | 127 (36.4) | 0.23 ± 0.21 | | 0.17 ± 0.17 | | 0.45 ± 0.24 | | 0.15 ± 0.14 | |
| Current shift type | | | .896 | | .832 | | .353 | | .204 |
| Day | 205 (58.7) | 0.22 ± 0.21 | | 0.16 ± 0.14 | | 0.42 ± 0.24 | | 0.19 ± 0.20 | |
| Evening | 106 (30.4) | 0.21 ± 0.18 | | 0.17 ± 0.15 | | 0.45 ± 0.24 | | 0.17 ± 0.16 | |
| Night | 38 (10.9) | 0.21 ± 0.21 | | 0.18 ± 0.21 | | 0.48 ± 0.25 | | 0.13 ± 0.10 | |
| Smoking habit | | | .736 | | .887 | | .489 | | .144 |
| No | 339 (97.1) | 0.22 ± 0.20 | | 0.17 ± 0.15 | | 0.44 ± 0.24 | | 0.17 ± 0.18 | |
| Yes | 10 (2.9) | 0.20 ± 0.21 | | 0.16 ± 0.11 | | 0.38 ± 0.26 | | 0.26 ± 0.28 | |
| Drinking habit | | | .498 | | .949 | | .658 | | .138 |
| No | 230 (65.9) | 0.23 ± 0.21 | | 0.17 ± 0.15 | | 0.44 ± 0.24 | | 0.16 ± 0.16 | |
| Yes | 119 (34.1) | 0.21 ± 0.19 | | 0.17 ± 0.15 | | 0.43 ± 0.24 | | 0.19 ± 0.22 | |
| Coffee-drinking habit | | | .238 | | .776 | | .042 | | .582 |
| No | 130 (37.2) | 0.21 ± 0.18 | | 0.16 ± 0.13 | | 0.46 ± 0.24 | | 0.17 ± 0.18 | |
| Occasionally | 134 (38.4) | 0.24 ± 0.23 | | 0.17 ± 0.14 | | 0.40 ± 0.23 | | 0.19 ± 0.19 | |
| At least 1 cup a day | 85 (24.4) | 0.20 ± 0.19 | | 0.17 ± 0.19 | | 0.47 ± 0.25 | | 0.16 ± 0.17 | |
| Exercise habit | | | .002 | | .241 | | .137 | | .581 |
| Almost never | 199 (57.0) | 0.25 ± 0.22 | | 0.16 ± 0.15 | | 0.42 ± 0.24 | | 0.17 ± 0.19 | |
| At least once a week | 150 (43.0) | 0.18 ± 0.16 | | 0.18 ± 0.15 | | 0.46 ± 0.24 | | 0.18 ± 0.17 | |

Abbreviations: *M*: mean; *SD*: standard deviation; ICU: Intensive Care Unit.

## Correlation between physical symptoms or health conditions and emotion recognition results in nurses

As shown in Table 2, anger detected by the app was not significantly related to migraines, peptic ulcers, or menstrual cycle patterns. However, nurses with peptic ulcers showed significantly higher levels of fear compared to those without the condition ($t=2.63$, $p=.014$). Regarding menstrual cycle regularity, female nurses with irregular cycles reported significantly lower levels of happiness ($t=-4.14$, $p<.001$) and higher levels of sadness ($t=6.22$, $p<.001$). These findings suggest that in the studied sample of nurses, irregular menstrual cycles were associated with reduced positive emotions and increased negative emotions.

## Correlation between app emotion recognition and validity measures

As shown in Table 3, anger was negatively correlated with PANAS-positive ($r=-.27$, $p<.001$) and OHQ ($r=-.25$, $p<.001$) but positively correlated with NAI ($r=.42$, $p<.001$). Fear correlated positively with stress (PSS, $r=.41$, $p<.001$), whereas happiness correlated positively with OHQ ($r=.45$, $p<.001$) and negatively with depression (BDI-II, $r=-.38$, $p<.001$). Sadness correlated positively with BDI-II ($r=.47$, $p<.001$) and negatively with OHQ ($r=-.26$, $p<.001$). These findings indicate that the app's recognition of emotions aligns with corresponding psychological constructs.

**Table 2. Correlation between physical symptoms or diseases and emotion recognition results of app (*N*=349).**

| Symptoms or diseases | Total (%) | Anger | | Fear | | Happiness | | Sadness | |
|---|---|---|---|---|---|---|---|---|---|
| | | *M±SD* | *p* | *M±SD* | *p* | *M±SD* | *p* | *M±SD* | *p* |
| Migraines | | | .220 | | .600 | | .878 | | .133 |
| Yes | 34 (9.7) | 0.26±0.22 | | 0.18±0.17 | | 0.43±0.21 | | 0.13±0.10 | |
| No | 315 (90.3) | 0.22±0.20 | | 0.17±0.15 | | 0.44±0.24 | | 0.18±0.19 | |
| Peptic ulcers | | | .105 | | .014 | | .252 | | .385 |
| Yes | 29 (8.3) | 0.16±0.14 | | 0.30±0.30 | | 0.39±0.25 | | 0.15±0.19 | |
| No | 320 (91.7) | 0.23±0.20 | | 0.16±0.12 | | 0.44±0.24 | | 0.18±0.18 | |
| Menstrual cycles* | | | .422 | | .073 | | <.001 | | <.001 |
| Irregular | 78 (22.3) | 0.21±0.16 | | 0.14±0.11 | | 0.34±0.21 | | 0.31±0.26 | |
| Regular | 220 (77.7) | 0.23±0.22 | | 0.18±0.16 | | 0.46±0.24 | | 0.13±0.12 | |

Abbreviations: *M*: mean; *SD*: standard deviation.

*Analysis only included the 298 female nurses.

**Table 3. Coefficients of correlation between emotion recognition results of app and validity criteria (*N*=349).**

| Validity criteria | Emotion recognition results | | | | | | | |
|---|---|---|---|---|---|---|---|---|
| | Anger | | Fear | | Happiness | | Sadness | |
| | *r* | *p* | *r* | *p* | *r* | *p* | *r* | *p* |
| Positive affect score of PANAS | −.27 | <.001 | −.03 | .609 | .31 | <.001 | −.09 | .105 |
| Negative affect score of PANAS | .12 | .023 | .13 | .014 | −.29 | <.001 | .14 | .009 |
| Mean score of OHQ | −.25 | <.001 | −.08 | .145 | .45 | <.001 | −.26 | <.001 |
| PSS score | −.16 | .003 | .41 | <.001 | −.02 | .701 | −.15 | .005 |
| BDI-II score | .06 | .291 | −.02 | .780 | −.38 | <.001 | .47 | <.001 |
| NAI-25 score | .42 | <.001 | −.11 | .051 | −.15 | .004 | −.16 | .003 |

Abbreviations: r: correlation coefficient; PANAS: Positive and Negative Affect Schedule; OHQ: Oxford Happiness Questionnaire; PSS: Perceived Stress Scale; BDI-II: Beck Depression Inventory-II; NAI-25: Novaco Anger Inventory-Short Form.

## Explanatory model of emotion recognition app after controlling confounding factors

As shown in Table 4, to clarify the factors influencing the app's emotion recognition results, multivariate analyses were performed using background and psychological variables from Tables 1–3. Confounders were selected based on univariate p values to avoid model overfitting. The models demonstrated significant explanatory power ($R^2 = .21 - .35$, all $p < .001$), and all VIFs were below 2.5, indicating no multicollinearity.

In the anger model, almost never exercising ($\beta = .11$, p = .025), lower PANAS-positive ($\beta = -.15$, p = .007), lower PSS ($\beta = -.13$, p = .006), and higher NAI ($\beta = .37$, p < .001) were significant. In the fear model, peptic ulcers ($\beta = .19$, p < .001) and higher PSS ($\beta = .36$, p < .001) were significant. In the happiness model (female), daily coffee ($\beta = .11$, p = .041), irregular menstrual cycles ($\beta = -.13$, p = .014), higher PANAS-positive ($\beta = .17$, p = .006), lower PANAS-negative ($\beta = -.14$, p = .034), and lower BDI ($\beta = -.14$, p = .034) were significant. In the sadness model (female), irregular menstrual cycles ($\beta = .30$, p < .001), lower PSS ($\beta = -.11$, p = .031), higher BDI ($\beta = .40$, p < .001), and lower NAI ($\beta = -.14$, p = .003) were significant.

## Test-retest reliability of emotion recognition results and validity measures

As shown in Table 5, all correlations between the app's emotion recognition results and the validity measures showed significant positive relationships in both pre- and post-tests, with correlation coefficients ranging from .56 to .90. This indicates a moderate to strong association. For test-retest reliability, the intraclass correlation coefficients (ICCs) for fear ranged from 0.50 to 0.75, reflecting moderate consistency. In contrast, the ICCs for anger, happiness, and sadness ranged from 0.75 to 0.90, indicating good consistency. Regarding the validity measures, the ICCs for PANAS positive affect, OHQ, PSS, and NAI scores also fell within the moderate range (0.50–0.75). Meanwhile, PANAS negative affect and BDI scores showed good consistency, with ICCs ranging from 0.75 to 0.90.

Table 4. Multiple linear regression analysis of factors related to emotion recognition results from the app.

| Independent variables | Anger (N=349) | | | | Fear (N=349) | | | | Happiness (Females only n=298) | | | | Sadness (Females only n=298) | | | |
|---|---|---|---|---|---|---|---|---|---|---|---|---|---|---|---|---|
| | β | SE | p | VIF | β | SE | p | VIF | β | S.E. | p | VIF | β | S.E. | p | VIF |
| Almost never exercising[a] | .11 | 0.02 | .025 | 1.03 | | | | | | | | | | | | |
| One or more cups of coffee a day[b] | | | | | | | | | .11 | 0.03 | .041 | 1.03 | | | | |
| Peptic ulcers[c] | | | | | .19 | 0.03 | <.001 | 1.06 | | | | | | | | |
| Irregular menstrual cycles[d] | | | | | | | | | −.13 | 0.03 | .014 | 1.16 | .30 | 0.21 | <.001 | 1.17 |
| PANAS positive affect score | −.15 | 0.002 | .007 | 1.46 | | | | | .17 | 0.002 | .006 | 1.52 | | | | |
| PANAS negative affect score | .10 | 0.002 | .080 | 1.57 | .03 | 0.001 | .571 | 1.07 | −.14 | 0.002 | .034 | 1.62 | −.01 | 0.002 | .929 | 1.57 |
| Mean OHQ score | −.12 | 0.02 | .083 | 2.13 | | | | | .15 | 0.03 | .054 | 2.29 | .02 | 0.21 | .685 | 1.64 |
| PSS score | −.13 | 0.002 | .006 | 1.11 | .36 | 0.001 | <.001 | 1.08 | | | | | −.11 | 0.002 | .031 | 1.09 |
| BDI score | −.09 | 0.001 | .132 | 1.61 | | | | | −.14 | 0.002 | .034 | 1.75 | .40 | 0.001 | <.001 | 1.73 |
| NAI score | .37 | 0.001 | <.001 | 1.03 | −.08 | 0.001 | .083 | 1.01 | | | | | −.14 | 0.001 | .003 | 1.02 |
| R² | .27 | | | | .21 | | | | .27 | | | | .35 | | | |
| F | 18.20 | | | | 23.10 | | | | 14.99 | | | | 26.34 | | | |
| p* | <.001 | | | | <.001 | | | | <.001 | | | | <.001 | | | |

Note: [a]Reference group: exercising at least once a week; [b]Reference group: occasional or no coffee; [c]Reference group: no peptic ulcers; [d]Reference group: regular menstrual cycles; *F-test of overall significance.

Abbreviations: β, standardized coefficients, SE, stands for the standard error, VIF, variance inflation factor; PANAS: Positive and Negative Affect Schedule; OHQ: Oxford Happiness Questionnaire; PSS: Perceived Stress Scale; BDI-II: Beck Depression Inventory-II; NAI-25: Novaco Anger Inventory-Short Form. $R^2$, overall R square value.

**Table 5. Test-retest reliability of emotion recognition results of app and validity criteria.**

| Study variables | Test phase | | Correlation | | ICC | 95% CI | |
|---|---|---|---|---|---|---|---|
| | Pre-test M±SD | Post-test M±SD | r | p | | Lower bound | Upper bound |
| Emotion recognition results | | | | | | | |
| Anger | 0.23±0.24 | 0.21±0.22 | .77 | <.001 | .76 | .56 | .88 |
| Fear | 0.18±0.13 | 0.17±0.11 | .73 | <.001 | .73 | .51 | .86 |
| Happiness | 0.47±0.22 | 0.50±0.22 | .78 | <.001 | .78 | .58 | .89 |
| Sadness | 0.12±0.10 | 0.13±0.09 | .80 | <.001 | .80 | .62 | .90 |
| Validity criteria | | | | | | | |
| Positive affect score of PANAS | 29.83±5.09 | 30.63±5.79 | .56 | .001 | .56 | .25 | .76 |
| Negative affect score of PANAS | 22.27±6.25 | 23.30±5.56 | .77 | <.001 | .77 | .57 | .88 |
| Mean score of OHQ | 3.79±0.62 | 3.91±0.66 | .70 | <.001 | .70 | .46 | .84 |
| PSS score | 25.43±5.72 | 25.97±5.17 | .60 | .001 | .60 | .29 | .78 |
| BDI-II score | 7.77±6.42 | 8.00±6.79 | .90 | <.001 | .90 | .80 | .95 |
| NAI-25 score | 45.50±17.13 | 47.00±16.38 | .69 | <.001 | .69 | .44 | .84 |

Abbreviations: *M*: mean; *SD*: standard deviation; ICC: intraclass correlation coefficient; 95% CI: 95% Confidence interval; PANAS: Positive and Negative Affect Schedule; OHQ: Oxford Happiness Questionnaire; PSS: Perceived Stress Scale; BDI-II: Beck Depression Inventory-II; NAI-25: Novaco Anger Inventory-Short Form.

## Discussion

The app detected four emotions: happiness was the most prominent; and anger, fear, and sadness were less intense. Nurses who exercised less showed more anger, those with peptic ulcers had higher fear, and female nurses with irregular menstrual cycles reported lower happiness and higher sadness. Multiple regression identified significant associations between these factors and app-detected emotions, suggesting preliminary psychological validity and the explanatory potential of the model.

A 2024 survey by the Taiwan Union of Nurses Association reported that the average age of nurses in Taiwan is 39, and 95.4% are women [36]. In contrast, our sample was younger (mostly aged 25–29). Participants who were pregnant or taking hormonal medications were excluded to reduce confounding. Consequently, the sample was predominantly young and female, differing from the national nursing workforce and possibly affecting external validity. Future studies should recruit nurses of broader age ranges and physiological conditions to enhance generalizability.

Relatively small regression coefficients ($\beta = .11$ or.13) nevertheless reflect the subtle but consistent influences of lifestyle and health factors on emotional states. In occupational health contexts, even small effect sizes could hold practical significance when accumulated across large nursing populations, suggesting potential value for preventive or wellness-oriented interventions.

The emotion recognition results from the app showed moderate positive correlations with standardized psychological scales (NAI, OHQ, PSS, and BDI), indicating its convergent validity. Test-retest analysis also demonstrated good consistency in recognizing anger, happiness, and sadness over time. As the AI VER app classifies emotions from voice features rather than from multiple questionnaire items, traditional analyses such as factor analysis or internal consistency tests (e.g., Cronbach's α or split-half reliability) were not applicable. This study therefore focused on psychometric validation, test-retest reliability, and convergent validity, rather than predictive performance [37,38]. Metrics such as accuracy or area under the curve were not computed; however, future studies could include receiver operating characteristic curve-area under the curve analyses with larger datasets to further quantify predictive capability.

Happiness was the most frequently detected emotion, possibly reflecting nurses' efforts to maintain positivity at work [39]. Emotional intelligence helps nurses manage stress by recognizing and regulating their emotions, supporting mental health and work efficiency [40]. Maintaining positive emotions also enhances nursing quality and job satisfaction [41]. Anger was the second-most prominent emotion, due perhaps to workplace stressors [42]. Furthermore, nurses who did not exercise regularly showed higher anger levels, due possibly to the role of exercise in releasing endorphins that reduce negative affect [43]. Sadness and fear were less frequent, suggesting that these emotions may be suppressed in high-stress environments.

This study found that nurses with a history of peptic ulcers were associated with higher fear scores. Although Lee et al. [44] suggested that fear may increase gastric acid secretion and weaken mucosal defenses, contributing to ulcer risk, our cross-sectional design limits causal inference. Similarly, lower happiness and higher sadness scores could be related to irregular menstrual cycles in female nurses. Prior research indicates that emotional stress may disrupt the hypothalamic-pituitary-ovarian axis, affecting hormone balance and menstrual regularity [45]. However, due to the study's design, we cannot confirm causality. Future longitudinal or experimental studies are needed to clarify these relationships.

This study included all participants ($N = 349$) when analyzing physical symptoms such as migraines and peptic ulcers. However, the analysis of menstrual cycles was limited to female nurses ($n = 298$), as this is a female-specific physiological factor. Consequently, the findings cannot be generalized to nurses of other genders, and thus the physiological-emotional associations identified should be interpreted within this gender-specific context. To improve external validity, future studies should include participants of diverse genders to explore how their physiological and psychological factors affect work-related emotions and health outcomes, thereby addressing current gaps in gender-inclusive occupational health research [46].

Sultson et al. found that unstable negative emotions predicted depression and anxiety over seven days of self-reports [47]. Likewise, Feng et al. reported that chronic stress was positively linked to depression and anxiety, mediating its association with fatigue [48]. Consistent with these findings, the emotion recognition scores of the app showed moderate positive correlations with standardized psychological scales, suggesting that it may reflect users' emotional tendencies. This provides preliminary evidence of the potential utility of the app for assessing clinical nurses' emotional responses.

The AI VER app achieved about 70% accuracy in emotion recognition, indicating moderate performance and some misclassification bias. Future studies should explore higher-performing or hybrid models to improve reliability [49]. A decision tree classifier was used for its interpretability and clinical practicality, as more complex models (e.g., neural networks) may reduce transparency and acceptance [50]. This approach balances accuracy and interpretability. Although suitable for research and preliminary screening, the current accuracy is insufficient for clinical decision-making. Thus, the AI VER app should be regarded as an assistive tool for identifying emotional tendencies rather than as a diagnostic system.

This study found evidence suggesting that the emotion recognition app demonstrated acceptable test-retest reliability among nurses. Emotions such as anger, happiness, and sadness showed high consistency, while fear showed moderate consistency. The validity criteria (NAI, PSS, OHQ, and BDI) also showed moderate to high correlations with app-derived emotion scores, suggesting both convergent validity and temporal stability [51–53]. However, the retest sample size was small, and the ICC for fear was only moderate, suggesting that stability and precision may be somewhat limited [54].

## Limitations

This study has several limitations. First, the small sample size may have affected statistical stability and result inference. Second, the cross-sectional design limits the ability to draw causal conclusions. Third, participants were recruited from a single medical institution using convenience sampling, which may have introduced selection bias and reduced external validity, thereby limiting generalizability to other regions or populations. In addition, physical health data such as exercise habits and medical history were self-reported, which potentially introduces recall bias or social desirability effects. Future research should consider multi-center and longitudinal designs to enhance both internal and external validity and to further verify the stability of these findings.

As this study was conducted in Taiwan, the results may have been influenced by cultural factors. In many Asian contexts, individuals tend to suppress negative emotions to conform to social norms, which could cause social desirability bias and affect both self-reports and app-based recognition [55]. Such cultural differences in emotional expression may limit the generalizability of the findings. Future studies should include cross-cultural samples to validate the tool's effectiveness [56].

## Conclusion

This study found that peptic ulcers and irregular menstrual cycles were associated with higher fear and sadness, and lower happiness and that nurses without regular exercise habits exhibited more anger. The AI VER app demonstrated acceptable validity and reliability, suggesting its potential utility as a supplementary tool for assessing nurses' emotional tendencies rather than as a definitive diagnostic instrument. This study focused on the measurement validity and correlation analysis of the AI VER app in nurses, without involving clinical referrals or mental health interventions. Although the app tentatively identified higher sadness scores among certain nurses, these findings should be interpreted as preliminary and exploratory. Future applications should consider ethical responsibilities and integrate with hospital mental health services to ensure timely support for individuals with high-risk emotions.

## Supporting information

**S1 File. Dataset: Anonymized background information of nurses, app-based emotion recognition data, and questionnaire scale responses.**
(XLS)

## Acknowledgments

The authors would like to express their sincere gratitude to all the nursing staff who participated in this study for their valuable time, commitment, and contributions. Their support was essential to the successful completion of this research.

## Author contributions

**Conceptualization:** Chu-Ying Huang.

**Formal analysis:** Wen-Pei Chang.

**Funding acquisition:** Wen-Pei Chang.

**Investigation:** Wen-Pei Chang.

**Methodology:** Wen-Pei Chang.

**Project administration:** Wen-Pei Chang.

**Resources:** Wen-Pei Chang.

**Software:** Wen-Pei Chang.

**Supervision:** Wen-Pei Chang.

**Writing – review & editing:** Wen-Pei Chang.

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
