## [Decision Letter · Decision Letter 0]

22 May 2025

Dear Dr. Chang,

We look forward to receiving your revised manuscript.

Kind regards,

Manuela Mendonça Figueirêdo Coelho, Ph.D

Academic Editor

PLOS ONE

Journal Requirements:

3. Thank you for stating the following financial disclosure: [This research was funded through an industry-academia collaboration agreement between Shuang Ho Hospital, Taipei Medical University and Bamboo Technology Co., Ltd. in Taiwan, with the project code A-112-006-S.]. 

4. Thank you for stating the following in the Acknowledgments Section of your manuscript: [This research was funded through an industry-academia collaboration agreement between Shuang Ho Hospital, Taipei Medical University and Bamboo Technology Co., Ltd. in Taiwan, with the project code A-112-006-S.]

Please remove any funding-related text from the manuscript and let us know how you would like to update your Funding Statement. Currently, your Funding Statement reads as follows: [This research was funded through an industry-academia collaboration agreement between Shuang Ho Hospital, Taipei Medical University and Bamboo Technology Co., Ltd. in Taiwan, with the project code A-112-006-S.]. 

5. In the online submission form, you indicated that [The data that support the findings of this study are available on request from the corresponding author. The data are not publicly available due to privacy or ethical restrictions.].

6. We note that Figure 1 in your submission contain copyrighted images. All PLOS content is published under the Creative Commons Attribution License (CC BY 4.0), which means that the manuscript, images, and Supporting Information files will be freely available online, and any third party is permitted to access, download, copy, distribute, and use these materials in any way, even commercially, with proper attribution. For more information, see our copyright guidelines: http://journals.plos.org/plosone/s/licenses-and-copyright.

Reviewers' comments:

Reviewer's Responses to Questions

**Comments to the Author**

1. Is the manuscript technically sound, and do the data support the conclusions?

Reviewer #1: Yes

Reviewer #2: Yes

2. Has the statistical analysis been performed appropriately and rigorously?

Reviewer #1: Yes

Reviewer #2: Yes

3. Have the authors made all data underlying the findings in their manuscript fully available?

Reviewer #1: Yes

Reviewer #2: No

4. Is the manuscript presented in an intelligible fashion and written in standard English?

Reviewer #1: Yes

Reviewer #2: Yes

Reviewer #1: Reviewer Comments

1. Ethical Considerations:

The manuscript mentions IRB approval and informed consent but lacks details on data anonymization and voice data protection. Given the sensitivity of biometric data, clarify how voice recordings were stored, processed, and anonymized to ensure participant confidentiality.

2. Sample Representation:

While the sample is predominantly female (85.4%), analyses involving menstrual cycles apply only to female participants. The manuscript should explicitly state whether male nurses were excluded from these specific analyses and discuss the implications for generalizability.

3. Measurement Validity:

The AI VER app’s accuracy (70%) is moderate. Discuss limitations arising from this accuracy level (e.g., potential misclassification bias) and its impact on results.

Justify the choice of a decision tree classifier over more advanced models (e.g., neural networks) for emotion classification, addressing trade-offs between interpretability and performance.

4. Causality vs. Correlation:

The discussion occasionally implies causation (e.g., fear leading to peptic ulcers). Clarify that the study identifies associations, not causality, and avoid overinterpretation of mechanisms without longitudinal or experimental evidence.

5. Confounding Variables:

Key confounders like workload intensity, job-specific stressors, or comorbid mental health conditions (e.g., anxiety) are not controlled. Acknowledge this limitation and suggest future studies to account for these variables.

6. Test-Rest Reliability:

The retest sample size (n=30) is small. While a 15% loss rate is mentioned, consider expanding the retest cohort to enhance reliability estimates, particularly for fear (moderate ICC).

7. Cultural Context:

The study was conducted in Taiwan. Discuss potential cultural influences on emotion expression (e.g., social desirability bias) and how these might affect the app’s performance or generalizability to other populations.

8. Data Availability:

PLOS ONE mandates public data availability unless restricted by ethical/legal concerns. The current statement (“available upon request”) may not comply. Either deposit data in a repository with access controls or provide a detailed justification for restrictions.

9. Figures and Tables:

Figure 1 is referenced but not included in the manuscript draft. Ensure all figures are properly formatted and captioned.

Table 1’s title (“three patient groups”) is inaccurate (participants are nurses). Revise for clarity.

10. Language and Clarity:

Several sentences are grammatically awkward or unclear (e.g., “The scores of the four emotions are A (anger), B (fear), C (happiness), and D (sadness)…”). Professional editing for language fluency is recommended.

11. Conflict of Interest:

The VER model was developed by Bamboo Technology Co., Ltd., which funded the study. While the authors declare no competing interests, explicitly address potential biases (e.g., whether the company influenced data analysis/interpretation).

12. Methodological Rigor:

The app’s training data (RAVDESS and “members”) lacks specificity. Describe the demographic composition of training datasets to assess model generalizability.

For menstrual cycle assessment, clarify how “irregularity” was operationalized (self-report vs. clinical criteria) to ensure validity.

Recommendation:

The study addresses an important topic with practical implications for nurse well-being. However, revisions addressing the above points are essential to strengthen validity, clarity, and compliance with journal guidelines. With these improvements, the manuscript would make a valuable contribution to the field.

Reviewer #2: The growing burden of mental health issues among healthcare professionals represents a global challenge. Health systems exert significant structural pressures on professionals, including nurses. While there is often a tendency to focus solely on individual determinants, it is essential to consider environmental and systemic factors, particularly workplace-related determinants, that influence mental health. Recognizing this broader context allows for a more comprehensive understanding: addressing mental health issues requires not only individual-level support but also systemic changes to the environments that contribute to psychological distress. This holistic approach is a major contemporary challenge for improving health outcomes and could be a little bit more addressed in the discussion part of your work.

The study could benefit from numerous revisions to enhance methodological rigor and strengthen the discussion (while taking into account limitations) in order to effectively disseminate its findings.

#Abstract:

Objectives: “AI” should be fully written line 21 as it is the first time you introduce it.

Results: Describe what kind of “exercise” you are talking about. Is it a physical exercise, a professional exercise?

Conclusion: Line 36 the word “that” is useless.

#Introduction:

Line 54 you are referring to VER with a paper that you cite to highlight that the VER could be used in multiple fields such as screening for depression. When I read the paper you cited (4th reference), I could not find in the paper an application of VER except for Alzheimer’s disease. I feel like this article was more about artificial intelligence techniques that could diagnose numerous diseases but not with the speech detection in general. The way you used that reference could be misleading and you should correct that.

Line 60, prefer the term “socio-demographic” rather than “personal”.

Line 61 for your mention that “studies have found gradually increasing work pressure on nurses.”, please provide references accordingly.

In line 66, since no evidence-based references support this statement, I would suggest using the term “could” instead of “can”.

Line 68: Please rewrite the sentence. “developed based” is grammatically incorrect.

The introduction section is generally well structured but may be somewhat too concise. It could benefit from additional contextual information to provide a more comprehensive background on the topic.

#Methods:

Line 79: “study subjects” � “study participants”

Please indicate in this section how many researchers were recording the nurse’s voice.

Please provide a translated questionnaire that nurses were answering.

Line 86: You mention that nurses recorded their voice for 60 seconds. However, in Figure 1 we can observe a length of conversation that appears to last 1 minutes and 30 seconds. Is 60 seconds a minimum required? Please clarify this. Also, could you please be more precise on what the nurse is supposed to talk about. No information is given on the content / topic of what the nurse is saying.

Line 91, 1st sentence: This should be included in the results section.

Line 91: Please make clearer the sentence for which you cite citation number 12 in which there is no data specifically on “emotion measurement tools” in this paper. Develop this aspect.

Ethical considerations section: please write it in past tense.

Line 109 “Research instruments”: Please introduce this section properly.

Line 111: What was the rationale for collecting data on religion? Is there any scientific literature related to the study topic that justifies its inclusion? It is important to ensure that all collected data comply with international ethical standards, particularly regarding confidentiality and relevance.

Line 112: What do you mean by “living habits” or “work experience”? Please be more precise on the variables you collected. Indeed, it contrasts with the level of detail you provided for the scores in the research instruments section.

Line 197: Please specify how much data you tested the model with and provide more precise data on its results.

Line 207: You mention using t-tests. Detail the conditions under which this test was applied: was the normality of the variable distributions verified for this parametric test?

#Results

The results related to background characteristics should be compared in the discussion section with national statistics on nurses to support the external validity of your findings.

Table 1 (line 232): Could you clarify the choice of this title? The reference to “three groups” is unclear, and the use of the term “patient” seems inconsistent with your study population, which consists solely of healthcare professionals (nurses). Please provide a justification for this wording.

Tables: Please add in the note part (or title) the total number of participants included for these data. Keep a homogeneous formatting for this information across tables.

Line 254: Consider adding “in this study population” or “among nurses” to avoid overgeneralization of the findings.

Line 273: Please confirm the wording “negative automatic thoughts”.

#Discussion

This section needs major revision.

While this section effectively presents the key results and relates them to existing literature, it lacks a critical analysis of the study. Notably, no limitations are discussed, despite several being apparent. For a more robust scientific discussion, a dedicated “Limitations” section should be added. You should consider addressing biases inherent to your study design, such as the small sample size, recall bias related to the use of self-administered questionnaires (especially for reporting physical symptoms or past medical history), and any other relevant methodological constraints. Clearly acknowledging these limitations will strengthen the transparency and credibility of your findings. You should clearly state some limitations with a discussion of internal and external validity.

#General comments:

If the AI VER app detects and reports a high sadness score, are there specific referral procedures in place to connect the nurse with a mental health professional?

Additionally, your paper does not clearly explain how the various types of collected data were processed. Readers may be left with the impression that a significant amount of data was gathered without a clear account of how each type was analyzed or utilized. The clarity of your work would benefit from a more detailed explanation of your data collection methods and the subsequent data processing steps. This would enhance the transparency and reproducibility of your study.

Thank you for your work on that interesting and important topic!

**Do you want your identity to be public for this peer review?** For information about this choice, including consent withdrawal, please see our Privacy Policy

Reviewer #1: No

Reviewer #2: **Yes: ** Matthieu Lebrat

---

## [Author Response · Author response to Decision Letter 1]

12 Jun 2025

PLOS ONE

Referee Report on the Manuscript ID PONE-D-25-20092

“Reliability and validity analysis of AI voice emotion recognition app and correlation between its results and the background variables and physical symptoms or diseases of nurses”

We are grateful for your invaluable assistance in the revision of this paper. For this revision, we have closely followed the guidance provided by the editor and reviewers.

Detailed responses to each of the editor’s and reviewers’ comments have been provided below, with reviewers’ comments, followed by our comments and revisions.

In accordance with Editor’s suggestions:

Response:

We thank the Editor for the reminder. We have ensured that our manuscript meets PLOS ONE’s style requirements, including those for file naming. We have referred to the official templates provided for the style examples. (see the Title page and Text)

Response:

We understand PLOS ONE’s guidelines on code sharing for submissions involving author-generated code. However, the artificial intelligence (AI) voice emotion recognition (VER) algorithm used and developed in this study is legally under trade secret protection. Due to intellectual property and trade secret considerations, we are unable to disclose any details regarding the algorithm content, source code, or internal computational mechanisms. However, we confirm that the data analysis, results, and conclusions of this study are all based on the output of the validated VER system and do not involve the disclosure of or reliance on algorithm details. We are committed to adhering to relevant regulations and properly protecting proprietary technologies and trade secrets while ensuring the scientific rigor and reproducibility of the research.

3. Thank you for stating the following financial disclosure: [This research was funded through an industry-academia collaboration agreement between Shuang Ho Hospital, Taipei Medical University and Bamboo Technology Co., Ltd. in Taiwan, with the project code A-112-006-S.].

Response:

We thank the Editor for the reminder. Please update our information in the online submission form. We have also added the following paragraph to the cover letter: "This research was funded through an industry-academia collaboration agreement between Shuang Ho Hospital, Taipei Medical University, and Bamboo Technology Co., Ltd. in Taiwan, with the project code A-112-006-S. The funders had no role in the study design, data collection and analysis, decision to publish, or preparation of the manuscript. The first author, Chu-Ying Huang, is an employee of Bamboo Technology Co., Ltd. and contributed only to the conceptualization of the study. All other aspects of the research design, data collection and analysis, and manuscript preparation were solely carried out by the corresponding author, Wen-Pei Chang." (see the Cover letter)

4. Thank you for stating the following in the Acknowledgments Section of your manuscript: [This research was funded through an industry-academia collaboration agreement between Shuang Ho Hospital, Taipei Medical University and Bamboo Technology Co., Ltd. in Taiwan, with the project code A-112-006-S.]

Please remove any funding-related text from the manuscript and let us know how you would like to update your Funding Statement. Currently, your Funding Statement reads as follows: [This research was funded through an industry-academia collaboration agreement between Shuang Ho Hospital, Taipei Medical University and Bamboo Technology Co., Ltd. in Taiwan, with the project code A-112-006-S.].

Response:

We thank the Editor for this reminder. We understand that funding information should not appear in the Acknowledgments section and should only be placed in the Funding Statement section of the online submission form. We have removed all funding-related text from the manuscript. Please update our information in the online submission form. We have also added the following paragraph to the cover letter: "This research was funded through an industry-academia collaboration agreement between Shuang Ho Hospital, Taipei Medical University, and Bamboo Technology Co., Ltd. in Taiwan, with the project code A-112-006-S. The funders had no role in the study design, data collection and analysis, decision to publish, or preparation of the manuscript. The first author, Chu-Ying Huang, is an employee of Bamboo Technology Co., Ltd. and contributed only to the conceptualization of the study. All other aspects of the research design, data collection and analysis, and manuscript preparation were solely carried out by the corresponding author, Wen-Pei Chang." (see the Cover letter)

5. In the online submission form, you indicated that [The data that support the findings of this study are available on request from the corresponding author. The data are not publicly available due to privacy or ethical restrictions.].

Response:

We thank the Editor for this reminder. We understand PLOS ONE’s open data policy. In accordance with this policy, we have uploaded an Excel file of the raw data as supplementary information for other researchers to review and reproduce the study results. (see the S1 Dataset)

6. We note that Figure 1 in your submission contain copyrighted images. All PLOS content is published under the Creative Commons Attribution License (CC BY 4.0), which means that the manuscript, images, and Supporting Information files will be freely available online, and any third party is permitted to access, download, copy, distribute, and use these materials in any way, even commercially, with proper attribution. For more information, see our copyright guidelines: http://journals.plos.org/plosone/s/licenses-and-copyright.

Response:

We thank the Editor for this reminder. We understand that all PLOS ONE content must be permitted for publication under the Creative Commons Attribution License (CC BY 4.0), and we have confirmed that the images in Fig 1 meet this requirement. We have sought permission from the original copyright holder of Fig 1 to publish the images in PLOS ONE under the CC BY 4.0 license, completed and uploaded the Content Permission Form, and added the following text to the caption of Fig 1: "(Image provided by Bamboo Technology Co., Ltd., published under a CC BY license, with permission from Bamboo Technology Co., Ltd., original copyright 2025.)" (see the Figure 1)

In accordance with Reviewer #1’s suggestions:

1. Ethical Considerations:

The manuscript mentions IRB approval and informed consent but lacks details on data anonymization and voice data protection. Given the sensitivity of biometric data, clarify how voice recordings were stored, processed, and anonymized to ensure participant confidentiality.

Response:

We thank the Reviewer for this reminder. We have added the following explanation to the Ethical considerations section: "To further protect the security of voice data and participant privacy, several data protection measures were implemented, including storing voice recordings on encrypted servers accessible only by authorized research team members using password protection. During data processing, all voice recordings were subjected to feature extraction using dedicated software, retaining only the acoustic parameters relevant to the analysis; the original voice files were not used or disclosed further. Any names or personally identifiable information contained in the recordings were deleted or anonymized prior to analysis, and the data were labeled with random codes; this completely removed any link to personal identities. All data were retained in accordance with the IRB-approved storage period and were securely destroyed after the retention period expired. The VER model used in the study was developed by Bamboo Technology Co., Ltd., which also funded the research. Although the company initiated the collaboration, it was not involved in data analysis, research processes, or decisions regarding study conclusions." (see the Ethical considerations Section on lines 126-138 on Page 8 and 9)

2. Sample Representation:

While the sample is predominantly female (85.4%), analyses involving menstrual cycles apply only to female participants. The manuscript should explicitly state whether male nurses were excluded from these specific analyses and discuss the implications for generalizability.

Response:

We thank the reviewer for the guidance. We have supplemented the following content in the Statistical analyses section: "As menstrual cycles are a physiological phenomenon unique to females, the menstrual cycle analysis was performed with only the data from female nurses. Male nurses were excluded from related analyses, and the results are only applicable to female nurses." (see the Statistical analyses Section on lines 280-283 on Page 16)

We have also added a note to the Menstrual cycles item in Table 2: "*Analysis only included the 298 female nurses." (see Table 2 on Page 26)

We have also added the following content to the Discussion section: "The analyses of specific physical symptoms (e.g., migraines and peptic ulcers) in this study included all of the participants (N = 349), that is, both male and female nurses. However, the analysis regarding menstrual cycles was limited to female nurses (n = 298); male participants were naturally excluded. As menstrual cycles are a female-specific physiological phenomenon, the findings cannot be directly generalized to male nurses or nurses of other genders. This analytical design limits the generalizability of the results across genders, particularly when exploring how physiological variables are related to emotions and physiological states. To enhance the external validity and applicability of future research, we recommend that future studies include participants of diverse genders or gender identities to examine the impact of their respective physiological and psychological variables on work-related emotions and health statuses, thus filling gaps in the existing body of research [46]." (see the Discussion Section on lines 440-450 on Page 36)

In response to this addition, we have cited the following study:

46. Polit FD, Beck CT. Nursing research: generating and assessing evidence for nursing practice. 11th Edition. Lippincott Williams & Wilkins; 2021.

3. Measurement Validity:

The AI VER app’s accuracy (70%) is moderate. Discuss limitations arising from this accuracy level (e.g., potential misclassification bias) and its impact on results.

Justify the choice of a decision tree classifier over more advanced models (e.g., neural networks) for emotion classification, addressing trade-offs between interpretability and performance.

Response:

We thank the reviewer for this reminder. We have supplemented the following content in the Research instruments section: "The preliminary model was trained using the RAVDESS data, which include 7,356 audio files of the voices of 24 professional actors (12 females and 12 males) between the ages of 21 and 40. All of these actors were native English speakers with North American accents. Their lines were delivered in various emotional states, which were then fine-tuned using a convolutional neural network classifier, achieving a preliminary accuracy of 79.2%." and "To enhance the appropriateness of the model for the expressions and culture of Mandarin speakers, the app collected voice data containing natural Mandarin spoken with a Taiwanese accent. The dataset contains 13,075 audio recordings submitted by 6,457 independent users and is 28,814 minutes in total. In the recordings, the participants talked freely and described their personal experiences and emotions. Three certified psychological counselors annotated all of the recordings using the same emotions used with RAVDESS. To process the subjectivity of the annotations, triangulation was adopted, and recordings with consistent annotations from at least two of the psychological counselors were retained. Ultimately, 1,961 training samples and 491 pen-and-paper test s

---

## [Decision Letter · Decision Letter 1]

18 Jul 2025

Dear Dr. Chang,

We look forward to receiving your revised manuscript.

Kind regards,

Helen Howard

Staff Editor

PLOS ONE

Journal Requirements:

**Additional Editor Comments:**

The manuscript has been evaluated by two reviewers, and their comments are available below.

The reviewers have raised a number of concerns that need attention. In particular, they request additional information on the AI tool, its validation, and the controlling for confounders. The reviewers also request that a copy of the questionnaire is provided as supplementary information.

Could you please revise the manuscript to carefully address the concerns raised?

Reviewers' comments:

Reviewer's Responses to Questions

**Comments to the Author**

Reviewer #1: All comments have been addressed

Reviewer #2: (No Response)

2. Is the manuscript technically sound, and do the data support the conclusions?

Reviewer #1: Yes

Reviewer #2: Yes

3. Has the statistical analysis been performed appropriately and rigorously?

Reviewer #1: Yes

Reviewer #2: Yes

4. Have the authors made all data underlying the findings in their manuscript fully available?

Reviewer #1: Yes

Reviewer #2: Yes

5. Is the manuscript presented in an intelligible fashion and written in standard English?

Reviewer #1: Yes

Reviewer #2: Yes

Reviewer #1: The study addresses an important topic with potential practical value. However, the critical lack of detail regarding the AI app and its validation, coupled with limitations in study design and analysis, currently undermine the conclusions. Addressing these concerns, particularly providing full transparency about the AI tool and its validation, and controlling for confounders, is essential for the manuscript to make a significant scientific contribution. The ethical handling of sensitive voice data must also be clearly demonstrated.

Reviewer #2: Thank you for your revision that significantly enhanced the manuscript quality.

I recommend to accept the paper after my last comment: Regarding point 10 that I raised (Reviewer 2). I do not see any questionnaire that the nurses filled out even though you answered that you will provide it. Is this an error ? It should be in Supplementary file.

King regards.

**Do you want your identity to be public for this peer review?** For information about this choice, including consent withdrawal, please see our Privacy Policy

Reviewer #1: No

Reviewer #2: **Yes: ** MATTHIEU LEBRAT

---

## [Author Response · Author response to Decision Letter 2]

4 Aug 2025

PLOS ONE

Editor-in-Chief

Aug 3, 2025

Dear Editor-in-Chief,

Thank you very much for giving us the opportunity to again revise our paper entitled “Reliability and validity analysis of AI voice emotion recognition app and correlation between its results with the background variables and physical symptoms or diseases of nurses” (Manuscript ID PONE-D-25-20092R1). We have completed the required revisions, closely following the suggestions of the editor and reviewers. You may find the details of these revisions in the attached files. We believe that we have fulfilled all of the requirements for submission of the paper to PLOS ONE. If you find that any details have been overlooked, or if you need any further information, please do not hesitate to contact me.

Yours sincerely,

Wen-Pei Chang, RN, PhD

PLOS ONE

Referee Report on the Manuscript ID PONE-D-25-20092R1

“Reliability and validity analysis of AI voice emotion recognition app and correlation between its results with the background variables and physical symptoms or diseases of nurses”

We are grateful for your invaluable assistance in the revision of this paper. For this revision, we have closely followed the guidance provided by the reviewers.

Detailed responses to each of the reviewers’ comments have been provided below, with reviewers’ comments, followed by our comments and revisions.

In accordance with Reviewer #1’s suggestions:

The study addresses an important topic with potential practical value. However, the critical lack of detail regarding the AI app and its validation, coupled with limitations in study design and analysis, currently undermine the conclusions. Addressing these concerns, particularly providing full transparency about the AI tool and its validation, and controlling for confounders, is essential for the manuscript to make a significant scientific contribution. The ethical handling of sensitive voice data must also be clearly demonstrated.

Response:

We thank the reviewer for their affirmation of the importance of the research topic and its potential practical value as well as for their concrete and valuable suggestions. As shown below, we have made revisions and supplemented content as suggested:

1.Regarding the data processing details of the emotion recognition model

We have supplemented the following explanation in the Research instruments section: "To enhance the performance of the model and cultural appropriateness for Mandarin Chinese speakers, the app initially collected natural Mandarin Chinese voice data from individuals with a Taiwanese accent. The dataset contains Mandarin Chinese voice data from users in Taiwan. A total of 6,457 participants took part since data collection began, producing 13,075 items of voice data. The recordings reached 28,814 minutes in total length, averaging approximately 2.2 minutes each. Furthermore, over 100,000 voice files have been accumulated in the historical data of the platform, showing scalability potential for future model training. All voice data underwent standard preprocessing, including noise reduction and automatic segmentation, to ensure audio quality and analysis consistency. The emotions in each item of voice data were labeled by three nationally certified counseling psychologists. The emotion label classifications included eight basic emotions: worried, happy, sad, angry, surprised, fearful, calm, and neutral. Multi-label emotion classifications were permitted to reflect the emotional complexity of actual voice expressions. To increase label consistency and reliability, we employed the triangulation method and only included data in which at least two of the psychologists assigned the same labels. Ultimately, 13,075 recordings were included. Using an 8:1:1 ratio, the recordings were divided into a training set (n = 10,460), a validation set (n = 1,307), and a test set (n = 1,308). A total of 1,961 training samples and 491 test samples were retained. The optimized Mandarin Chinese model had a 79.8% accuracy rate. The open participation mechanism of the app allowed for a wide range of natural voice variations, thereby enhancing the ecological validity of the model and reflecting real-world usage scenarios. Note that we did not systematically obtain structured demographic statistics during the data collection process, which prevented us from analyzing model performance from an ethnic aspect. Future studies could include such variables for more comprehensive assessments of model versatility." (see the Research instruments Section on lines 219-242 on Page 13-14)

2.Regarding the training and validation of the emotion recognition model

We have supplemented the following explanation in the Research instruments section: "The emotion recognition model in this study was based on the wav2vec2-Large pre-trained audio representation framework and fine-tuned to adapt to Mandarin Chinese voice emotion recognition tasks. The model was trained using the triangulation-validated voice data. After each epoch, the performance of the model was validated using the test set, which was not part of the training, to observe the learning progress and generalization ability of the model. The model displayed a steady learning trend during the initial phases of training, with accuracy steadily rising from 48.9% after the first epoch to 85.9% after the fifteenth epoch. The relatively stable performance of the model demonstrated that voice emotion characteristics have been effectively learned with good classification performance and consistency." (see the Research instruments Section on lines 243-252 on Page 14)

3.Regarding the control of confounders

We have added a section with the heading “Explanatory model of emotion recognition app after controlling confounding factors” and Table 4 to the Results, with the following content: "To clarify the factors that impact the emotion recognition results from the app and control potential confounding factors, we referred to the background variables and psychological scale results presented in Tables 1 through 3 and employed multivariate statistical methods to control potential confounding effects. Confounders were selected based on the p values of the univariate analysis results to prevent over-tuning the model. The regression analysis results in Table 4 show that the model possesses significant explanatory power with regard to the recognition results of the various emotions (R2 ranging from .212 to .352, p < . 001). The VIFs of the variables were also all less than 2.5, thereby indicating the absence of multicollinearity. This analysis provides a basis for inference after controlling confounders and enhances the reliability and explanatory power of this study in terms of the validity of the emotion recognition model.

In the anger model, almost never exercising (β = .11, p = .025), the PANAS positive affect score (β = -.15, p = .007), the PSS score (β = - .13, p = .006), and the NAI score (β = .37, p < .001) were significant factors. In the fear model, peptic ulcers (β = .19, p < .001) and the PSS score (β = .36, p < .001) were significant factors. In the happiness model for the female participants, one or more cups of coffee a day (β = .11, p = .041), irregular menstrual cycles (β = -.13, p = .014), the PANAS positive affect score (β = .17, p = .006), the PANAS negative affect score (β = -.14, p = .034), and the BDI score (β = -.14, p = .034) were significantly correlated. In the sadness model for the female participants, irregular menstrual cycles (β = .30, p < .001), the PSS score (β = -.11, p = .031), the BDI score (β = .40, p < .001), and the NAI score (β = -.14, p = .003) were significantly correlated. Thus, these regression results indicate that after personal and psychological variables were controlled, the app still demonstrated significant validity and explanatory power in recognizing different emotions." (see the Results Section on lines 356-379 on Page 29-30 and Table 4 on Page 31-32)

Furthermore, to maintain the consistency of the entire paper, we have added the following content to the abstract: "To control for potential confounding variables, demographic and health-related factors including age, gender, work experience, exercise habits, and history of physical symptoms were collected and statistically adjusted in correlation analyses." and "Multiple linear regression analysis revealed the following: (1) nurses without regular exercise exhibited more anger, while those with peptic ulcers reported greater fear; (2) drinking over one cup of coffee daily was linked to higher happiness; and (3) female nurses with irregular menstrual cycles had lower happiness and greater sadness." (see the Abstract on lines 22-36 on Page 2-3)

In addition, we have supplemented the following explanation in the Discussion: "Multiple linear regression further confirmed that several background and psychological factors were significantly associated with the emotions detected by the app, supporting the preliminary psychological validity and explanatory power of the model." (see the Discussion Section on lines 406-408 on Page 36)

4.Regarding the ethical handling of sensitive voice data

We have added the following explanation to the Ethical considerations section: "Users themselves could remove their voice data on the platform at any time. Furthermore, all data were stored on a server encrypted by Microsoft Azure and managed in accordance with IRB-approved security regulations to ensure the security and legality of data access and storage processes." (see the Ethical considerations Section on lines 130-133 on Page 8)

In accordance with Reviewer #2’s suggestions:

Thank you for your revision that significantly enhanced the manuscript quality.

I recommend to accept the paper after my last comment: Regarding point 10 that I raised (Reviewer 2). I do not see any questionnaire that the nurses filled out even though you answered that you will provide it. Is this an error ? It should be in Supplementary file.

King regards.

Response:

We thank the reviewer for their affirmation of our revisions. We have supplemented and uploaded the questionnaire as a supplementary file. We apologize for this oversight and thank you for the reminder and correction.

---

## [Decision Letter · Decision Letter 2]

19 Oct 2025

Dear Dr. Chang,

Thank you for submitting your manuscript to PLOS ONE. After careful consideration, we feel that it has merit but does not fully meet PLOS ONE’s publication criteria as it currently stands. Therefore, we invite you to submit a revised version of the manuscript that addresses the points raised during the review process.

We look forward to receiving your revised manuscript.

Kind regards,

Mikiyas Amare Getu

Academic Editor

PLOS ONE

Journal Requirements:

Additional Editor Comments:

The topic is innovative and highly relevant to the emerging field of digital emotion recognition in healthcare. However, there are several areas of weakness that may need revision before final acceptance:

1) The title is overly long; consider shortening to emphasize the primary focus

2) Abstract lacks detail on the psychometric methods (e.g., type of reliability coefficients, what “moderate and positive correlation” means numerically).

3) The AI system’s accuracy (≈70–80%) should be contextualized — was this sufficient for clinical applications?

4) Which validity do you mean? Criterion validity, convergent validity, or concurrent validity?

5) Regarding sample size: Convenience sampling in one hospital limits generalizability; this needs to be discussed as a limitation

6) No factor analysis was performed to assess construct validity. And no inter-rater or internal consistency analysis for the app (e.g., Cronbach’s α or split-half reliability). Justify it

7) Effect sizes are small (e.g., β = .11 or .13) and their practical significance is not discussed.

8) Discussion tends to overstate the implications — the conclusion that the app can “objectively detect emotional patterns” may be premature given moderate correlations and app accuracy.

9) Causal language should be avoided (e.g., “Peptic ulcers were associated with fear” → should be “Peptic ulcers correlated with fear”).

10) How does the app perform compared with self-reported emotion scales (e.g., accuracy, AUC)?

11) The limitation should include Single-institution sample bias: Self-report physical health data (potential recall bias).

12) Was there any independent validation to prevent bias from the company developing the app?

Reviewers' comments:

Reviewer's Responses to Questions

**Comments to the Author**

Reviewer #1: All comments have been addressed

2. Is the manuscript technically sound, and do the data support the conclusions?

Reviewer #1: Yes

3. Has the statistical analysis been performed appropriately and rigorously?

Reviewer #1: Yes

4. Have the authors made all data underlying the findings in their manuscript fully available?

Reviewer #1: Yes

5. Is the manuscript presented in an intelligible fashion and written in standard English?

Reviewer #1: Yes

Reviewer #1: The authors have addressed prior review comments satisfactorily, and the study adheres to ethical standards. No further concerns regarding dual publication or ethics.

**Do you want your identity to be public for this peer review?** For information about this choice, including consent withdrawal, please see our Privacy Policy

Reviewer #1: No

---

## [Author Response · Author response to Decision Letter 3]

2 Nov 2025

PLOS ONE

Editor-in-Chief

Nov 2, 2025

Dear Editor-in-Chief,

Thank you very much for giving us the opportunity to again revise our paper entitled “Reliability, validity, and correlates of an AI voice emotion recognition app among nurses” (Manuscript ID PONE-D-25-20092R2). We have completed the required revisions, closely following the suggestions of the editor and reviewers. You may find the details of these revisions in the attached files. We believe that we have fulfilled all of the requirements for submission of the paper to PLOS ONE. If you find that any details have been overlooked, or if you need any further information, please do not hesitate to contact me.

Yours sincerely,

Wen-Pei Chang, RN, PhD

PLOS ONE

Referee Report on the Manuscript ID PONE-D-25-20092R2

“Reliability, validity, and correlates of an AI voice emotion recognition app among nurses”

In accordance with Editor’s suggestions:

1.The title is overly long; consider shortening to emphasize the primary focus.

Response:

We have shortened the title to make it more concise and focused on the main aim of the study. The revised title is as follows: “Reliability, validity, and correlates of an AI voice emotion recognition app among nurses”.

This version highlights the psychometric evaluation as the primary focus while retaining the key contextual information regarding its use among nurses. The change has been applied to both the title page and the manuscript.

2.Abstract lacks detail on the psychometric methods (e.g., type of reliability coefficients, what “moderate and positive correlation” means numerically).

Response:

We have revised the Methods and Results sections of the Abstract to include detailed psychometric information, specifying the type of reliability coefficient (intraclass correlation coefficient, ICC) and the numerical values of the convergent validity correlations (Page 2-3, Abstract, highlighted in red). Specifically, Pearson’s correlations between app-derived emotions and corresponding standardized scales were reported (anger–Novaco Anger Inventory-Short Form, r = .42; fear–Perceived Stress Scale, r = .41; happiness–Oxford Happiness Questionnaire, r = .45; sadness–Beck Depression Inventory-II, r = .47; all p < .001), and test-retest reliability was evaluated using ICCs (range = 0.73-0.80).

3.The AI system’s accuracy (≈70–80%) should be contextualized — was this sufficient for clinical applications?

Response:

We have clarified in the Discussion section that although the AI VER app achieved approximately 70% accuracy, which is acceptable for research and preliminary screening purposes, this is not sufficient for direct clinical decision-making (Page 28, Discussion, highlighted in red). The system should therefore be regarded as a supplementary tool to assist in identifying emotional tendencies rather than a diagnostic system.

4.Which validity do you mean? Criterion validity, convergent validity, or concurrent validity?

Response:

We have clarified that the validity examined in this study refers to convergent validity, as the AI VER emotion scores were correlated with standardized psychological scales assessing similar emotional constructs (e.g., anger, fear, happiness, and sadness) (Page 2, Abstract; Page 13, Statistical Analyses; highlighted in red).

5.Regarding sample size: Convenience sampling in one hospital limits generalizability; this needs to be discussed as a limitation

Response:

We agree with the reviewer’s concern and have addressed this issue in the Discussion section (Page 29, Limitations, highlighted in red). Specifically, we have acknowledged that participants were recruited from a single medical institution using convenience sampling, which limits the external validity and generalizability of the findings to other settings or populations. This statement has been retained and slightly refined to ensure clarity.

6.No factor analysis was performed to assess construct validity. And no inter-rater or internal consistency analysis for the app (e.g., Cronbach’s α or split-half reliability). Justify it

Response:

The AI VER app is an algorithmic model that classifies emotions from acoustic features rather than a multi-item psychometric scale. Therefore, traditional factor analysis for construct validity and internal consistency indices (e.g., Cronbach’s α or split-half reliability) were not applicable. Instead, psychometric evaluation focused on test-retest reliability and convergent validity to assess temporal stability and consistency with standardized psychological measures. This clarification has been added to the Discussion section (Page 26, Discussion, highlighted in red).

7.Effect sizes are small (e.g., β = .11 or .13) and their practical significance is not discussed.

Response:

We have added an explanation in the Discussion section to interpret the small effect sizes in a practical context (Page 25-26, Discussion, highlighted in red). Although some standardized regression coefficients (β = .11 or .13) were small, they still indicate meaningful patterns in nurses’ emotional states, reflecting subtle but consistent influences of lifestyle and health factors (e.g., exercise habits and menstrual irregularity) on emotion recognition outcomes. In large populations, such small effects may accumulate and have practical implications for nurse well-being and workplace interventions.

8.Discussion tends to overstate the implications — the conclusion that the app can “objectively detect emotional patterns” may be premature given moderate correlations and app accuracy.

Response:

We have revised both the Discussion and Conclusion sections to adopt a more cautious and evidence-based tone (Page 25-29, Discussion; Page 30, Conclusion; highlighted in red). Specifically, we have replaced the statement that the app can “objectively detect emotional patterns” with phrasing that emphasizes its preliminary and assistive role. The revised text now indicates that the AI VER app may tentatively reflect emotional tendencies and should be regarded as a supplementary or exploratory tool, rather than a diagnostic system, given its moderate correlations and 70-80% accuracy. These revisions clarify that the current findings provide preliminary evidence of potential utility and that further validation in larger and more diverse samples is required.

9.Causal language should be avoided (e.g., “Peptic ulcers were associated with fear”→ should be “Peptic ulcers correlated with fear”).

Response:

We have carefully revised the related sentences in the Discussion section to avoid any causal wording (Page 30, Discussion, highlighted in red). Specifically, we have replaced expressions implying causation with neutral, correlational terms (e.g., “were associated with”) and explicitly stated that the cross-sectional design limits causal inference.

10.How does the app perform compared with self-reported emotion scales (e.g., accuracy, AUC)?

Response:

The present study focused on psychometric validation (reliability and validity) rather than predictive performance evaluation. We have added clarification in the Discussion stating that metrics such as accuracy or AUC were not computed but that future studies may include ROC-AUC analyses with larger datasets to further quantify predictive capability (Page 26, Discussion, highlighted in red).

11.The limitation should include Single-institution sample bias: Self-report physical health data (potential recall bias).

Response:

The single-institution sampling bias had already been addressed in the Limitations section. We have now added a statement noting that physical health data (e.g., exercise habits and medical history) were self-reported and may be subject to recall bias or social desirability effects (Page 29, Limitations, highlighted in red).

12.Was there any independent validation to prevent bias from the company developing the app?

Response:

We have clarified this point in the revised Ethical considerations section (Page 6, Ethical considerations, highlighted in red). Although the VER model was developed and funded by Bamboo Technology Co., Ltd., all data collection, statistical analyses, and result interpretations were conducted independently by our academic research team at Taipei Medical University to ensure objectivity and prevent developer bias. The company was not involved in data analysis, research procedures, or study conclusions.

---

## [Editor Report · Decision Letter 3]

7 Dec 2025

Reliability, validity, and correlates of an AI voice emotion recognition app among nurses

PONE-D-25-20092R3

Dear Dr. Wen-Pei Chang,

We’re pleased to inform you that your manuscript has been judged scientifically suitable for publication and will be formally accepted for publication once it meets all outstanding technical requirements.

Kind regards,

Mikiyas Amare Getu

Academic Editor

PLOS One

---

## [Editor Report · Acceptance letter]

PONE-D-25-20092R3

PLOS One

Dear Dr. Chang,

I'm pleased to inform you that your manuscript has been deemed suitable for publication in PLOS One. Congratulations! Your manuscript is now being handed over to our production team.

Kind regards,

on behalf of

Dr. Mikiyas Amare Getu

Academic Editor

PLOS One